# A Learn-to-Optimize Approach for Coordinate-Wise Step Sizes for Quasi-Newton Methods

## Abstract

Tuning step sizes is crucial for the stability and efficiency of optimization algorithms. While adaptive coordinate-wise step sizes have been shown to outperform scalar step size in first-order methods, their use in second-order methods is still under-explored and more challenging. Current approaches, including hypergradient descent and cutting plane methods, offer limited improvements or encounter difficulties in second-order contexts. To address these limitations, we first conduct a theoretical analysis within the Broyden-Fletcher-Goldfarb-Shanno (BFGS) framework, a prominent quasi-Newton method, and derive sufficient conditions for coordinate-wise step sizes that ensure convergence and stability. Building on this theoretical foundation, we introduce a novel learn-to-optimize (L2O) method that employs LSTM-based networks to learn optimal step sizes by leveraging insights from past optimization trajectories, while inherently respecting the derived theoretical guarantees. Extensive experiments demonstrate that our approach achieves substantial improvements over scalar step size methods and hypergradient descent-based method, offering up to $4\times$ faster convergence across diverse optimization tasks.

## 1 Introduction

Step size is an essential hyperparameter in optimization algorithms. It determines the rate at which the optimization variables are updated, and greatly influences the convergence speed and stability of the optimization process. In *first-order* gradient-based optimization, how to choose an appropriate step size is well studied: The step size is typically adjusted adaptively using past gradient information such as in AdaGrad Duchi et al. (2011), RMSProp Hinton (2012), and Adam Kingma (2015) for stochastic optimization tasks. These methods have demonstrated significant efficacy across a range of machine learning applications by dynamically tailoring the update scale for each iteration.

Step size in *second-order* methods received much less attention thus far. Second-order methods leverage the curvature information to adjust both the search direction and step size, offering faster convergence in number of iterations, at the cost of high computational complexity in calculating the Hessian (or its approximation) Wright (2006). A natural and common approach for step size tuning here is line search, which iteratively adjusts a *scalar* step size along the descent direction until certain conditions, such as the Armijo condition, are met Armijo (1966).

In contrast to scalar step size, we study the more general *coordinate-wise* step sizes (CWSS) in this work, which allow for individual variables to have different step sizes. CWSS are beneficial since different optimization variables may have different sensitivities to the step size; scalar step size is obviously a special case. They have also been shown to improve convergence in first-order methods Amid et al. (2022); Kunstner et al. (2023); Duchi et al. (2011).

In this work, we explore the impact of CWSS in the context of second-order methods, which remains largely unexplored to our knowledge. We choose the Broyden-Fletcher-Goldfarb-Shanno (BFGS) method Broyden (1965), one of the most widely used second-order optimization methods, as the backbone method. BFGS belongs to the quasi-Newton family of methods that iteratively update an approximation of the Hessian matrix using gradient information to reduce the complexity.

We start our study by demonstrating that existing solutions to tune CWSS in first-order methods do not work well in second-order contexts. The first such approach is hypergradient descent Maclaurin et al. (2015); Massé & Ollivier (2015), which iteratively tunes step sizes using their gradients at each BFGS step. We show empirically that it provides only marginal gains after the initial few steps of BFGS. Moreover, cutting-plane techniques, which expand backtracking line search into multiple dimensions, iteratively refine step sizes within feasible sets narrowed down by hypergradient-based incisions Kunstner et al. (2023). This method essentially offers an approximation of the Hessian in a first-order framework, thus complicating its direct application to second-order methods, in which Hessian approximation is handled by BFGS update, and the step sizes are adjusted to improve the Hessian approximation. Further, the intricate curvature within the Hessian presents additional challenges in plane cutting.

Therefore, we explore the learn-to-optimize (L2O) paradigm Andrychowicz et al. (2016) in this work. L2O replaces handcrafted rules with data-driven machine learning models that can adaptively learn efficient strategies, tailoring optimization processes to specific problem structures Andrychowicz et al. (2016); Lv et al. (2017). L2O has shown promising results in first-order optimization by predicting the optimal step sizes dynamically based on the current optimization state Liu et al. (2023); Song et al. (2024).

The application of L2O in quasi-Newton methods presents challenges. Whereas in first-order approaches, the step size primarily regulates the update magnitude, in second-order methods, it also affects the precision of Hessian approximations Wright (2006). This dual role adds complexities to step size tuning. Consequently, the unconstrained exploration inherent in conventional L2O makes convergence and stability harder to achieve within second-order L2O frameworks.

To address these challenges, we provide a theoretical analysis of coordinate-wise step sizes within the BFGS framework. We begin by outlining essential theoretical requirements for effective CWSS, aiming to ensure reliable optimization outcomes. These include achieving guaranteed convergence to a solution, maintaining stable progress towards the optimum, and preserving the strong convergence rates inherent to BFGS method. Guided by these foundational principles, we then derive a set of sufficient conditions for the CWSS matrix. They effectively define a "safe operating region", steering the learning process away from potentially unstable or divergent behaviours for better efficiency. While meeting these sufficient conditions ensures desirable properties like convergence, they do not determine the optimal strategy for fastest progress. Our L2O approach is therefore designed to learn the most effective step-size selection strategy within this theoretically defined safe region, leveraging insights from past optimization trajectories to accelerate performance.

Specifically, we propose a customized L2O model, featuring a LSTM network, to generate CWSS for BFGS method. Motivated by theoretical analysis, our model takes optimization variables, gradients, and second-order search directions as input. Distinct from many first-order L2O approaches that utilize longer unrolling horizons Liu et al. (2023), our model is trained with more frequent parameter updates to better capture the immediate effects of step size tuning in the sensitive quasi-Newton context. The training objective minimizes the expected objective value at the next iteration, augmented by a regularization term designed to ensure the learned step sizes adhere to our theoretical conditions for stability and efficient convergence.

We summarize our key contributions as follows:

1. We are the first to investigate coordinate-wise step size tuning in the context of second-order optimization methods, specifically the BFGS algorithm.

2. We establish theoretical foundation by deriving sufficient conditions for CWSS in the BFGS algorithm, ensuring convergence and stability and forming the principled basis for our L2O approach.

3. We propose a new L2O method to generate CWSS for the BFGS algorithm, integrating both theoretical principles and adaptive learning to guide the optimization process.

4. We empirically demonstrate the significant advantages of our method through extensive experiments on diverse optimization tasks, including classic optimization problems as well as a more challenging neural network training scenario. Our approach consistently achieves substantial speedups, delivering up to $4\times$ faster convergence when compared to classic backtracking line search and hypergradient descent methods. Notably, the performance ad-

vantage of our method typically becomes more pronounced as the problem dimensionality increases, highlighting its strong scalability. Furthermore, our method exhibits improved stability, evidenced by lower variance in performance across multiple runs.

## 2 PRELIMINARIES

In this chapter, we introduce the basics of second-order optimization methods, with a focus on BFGS. We show how step size tuning critically affects both the convergence and the quality of Hessian approximations. Then we establish the key assumptions that will support our analysis of CWSS in BFGS framework.

### 2.1 SECOND-ORDER METHODS

Second-order optimization methods, such as Newton's method Atkinson (1991), utilize both gradient and curvature information to find the minimum of an objective function. While first-order methods typically achieve a sub-linear convergence rate Beck (2017), second-order methods generally exhibit a faster, superlinear convergence rate Wright (2006). In Newton's method, the objective function is locally approximated by a quadratic function around the current parameter vector $x_k$:

$$g(y) \approx f(x_k) + \nabla f(x_k)^T(y - x_k) + \frac{\alpha_k}{2}(y - x_k)^T H_k(y - x_k), \tag{1}$$

where $H_k$ is the Hessian matrix and $\alpha_k$ is the damped parameter. By minimizing the quadratic approximation, the update rule for Newton's method becomes Wright (2006):

$$x_{k+1} = x_k - \alpha_k H_k^{-1} \nabla f(x_k). \tag{2}$$

Computing the Hessian is quite expensive and often infeasible for large-scale problems Pearlmutter (1994). Instead, quasi-Newton methods were proposed to approximate the Hessian to be more affordable and scalable Dennis & Moré (1977); Broyden (1967). Generally, quasi-Newton methods maintain an approximation of the Hessian matrix $B_k \approx H_k$ at each iteration, updating it with a rank one or rank two term based on the gradient differences between two consecutive iterations Conn et al. (1991); Broyden (1965). During this process, the Hessian approximation is restricted to follow the secant equation Wright (2006):

$$B_{k+1}s_k = y_k, \tag{3}$$

where $s_k = x_{k+1} - x_k$ and $y_k = \nabla f(x_{k+1}) - \nabla f(x_k)$. In the most common BFGS method, the Hessian approximation $B_k$ is updated at each iteration using the formula:

$$B_{k+1} = B_k - \frac{B_k s_k s_k^T B_k}{s_k^T B_k s_k} + \frac{y_k y_k^T}{y_k^T s_k}. \tag{4}$$

Although the enrollment of curvature information can greatly assist the optimization process, it also makes the algorithm more sensitive to the step size selection Wills & Schön (2018). The step size influences the update of the Hessian approximation, and an inappropriately large step can lead to violations of the curvature condition $y_k^T s_k > 0$, potentially resulting in an indefinite Hessian approximation Wright (2006). The step size must balance between exploiting the current curvature information (encoded in $B_k$) and allowing for sufficient exploration of the parameter space. This balance is more delicate than in first-order methods due to the adaptive nature of the search direction.

### 2.2 ASSUMPTIONS

Our objective is to minimize the convex objective function $f(x)$ over $x \in \mathbb{R}^n$: $\min_{x \in \mathbb{R}^n} f(x)$. Our analysis relies on the following standard assumptions regarding the objective function $f$ and the Hessian approximations $B_k$. These assumptions are common in optimization literature Song et al. (2024); Liu et al. (2023); Wright (2006):

**Assumption 1.** *The objective function $f$ is $L$-smooth, meaning there exists a constant $L$ such that:*

$$\|\nabla f(x) - \nabla f(y)\| \leq L\|x - y\|. \tag{5}$$

**Assumption 2.** *The gradient $\nabla f(x)$ is differentiable in an open, convex set $D$ in $\mathbb{R}^n$, and $\nabla^2 f(x)$ is continuous at the minimizer $x^*$ with $\nabla^2 f(x^*)$ being nonsingular.*

**Assumption 3.** *The Hessian approximation generated by BFGS method is positive definite. Furthermore, there exists a constant $M \geq 1$ such that:*

$$cond(B_k) = \lambda_{\max}(B_k)/\lambda_{\min}(B_k) \leq M, \tag{6}$$

*where $\lambda_{\min}(B_k)$ and $\lambda_{\max}(B_k)$ are the smallest and largest eigenvalues of $B_k$, respectively. By this assumption we assume the Hessian approximation remain well-conditioned.*

**Assumption 4.** *The norm of update direction $B_k^{-1}\nabla f(x_k)$ is upper bounded by a constant $R$:*

$$\|B_k^{-1}\nabla f(x_k)\| \leq R. \tag{7}$$

*This is a standard assumption in the analysis of quasi-Newton methods, as $B_k^{-1}$ is maintained bounded through stable Hessian approximations Broyden (1967), and gradients $\nabla f(x_k)$ typically diminish near optimal points, ensuring the update direction remains controlled.*

## 3 COORDINATE-WISE STEP SIZES FOR BFGS

In this section, we first analyze the theoretical advantages of CWSS and then explore hypergradient descent as a practical method for its tuning. However, the limited improvements achieved through hypergradient descent reveal the challenges of finding effective CWSS, prompting us to consider alternative approaches. We resort to L2O method that can directly learn the step sizes from data derived from similar optimization problems. Building on this perspective, we establish sufficient conditions for effective CWSS that ensure convergence and descent properties, thus laying a solid foundation for learning-based approaches that can predict optimal step sizes efficiently during optimization.

### 3.1 GAIN OF COORDINATE-WISE STEP SIZES

To illustrate the potential benefits of CWSS in the BFGS method, let us consider the theoretical implications of relaxing the constraint of scalar step size. Assume we have identified an optimal scalar step size, denoted by $\alpha_k^*$, for the $k$-th iteration. If we allow the step size to be a diagonal matrix $P_k$ rather than a scalar, the optimality condition of $\alpha_k^*$ may no longer hold. To explore this, we can set the coordinate-wise step sizes $P_k$ as:

$$P_k = \alpha_k^* I - \frac{1}{LR} v_k B_k^{-1}\nabla f(x_k), \tag{8}$$

where $v_k = \text{diag}(\nabla f(x_k - \alpha_k^* B_k^{-1}\nabla f(x_k)))$, $L$ is the Lipschitz constant of $\nabla f$ and $R$ is from Assumption 4. This coordinate-wise step size $P_k$ is theoretically guaranteed to perform better than the scalar step size $\alpha_k^*$:

$$f(x_k - P_k B_k^{-1}\nabla f(x_k)) \leq f(x_k - \alpha_k^* B_k^{-1}\nabla f(x_k))$$
$$- \frac{1}{2LR}|\nabla f(x_k - \alpha_k^* B_k^{-1}\nabla f(x_k)) \odot B_k^{-1}\nabla f(x_k)|^2. \tag{9}$$

This demonstrates that CWSS in the BFGS method can yield a more substantial decrease in the objective function than a scalar step size. A more detailed analysis is provided in Appendix C.

### 3.2 NUMERICAL ANALYSIS OF COORDINATE-WISE STEP SIZE: A HYPERGRADIENT DESCENT METHOD

Building upon section 3.1, we investigate hypergradient descent on coordinate-wise step size matrix $P_k$. The update rule with CWSS takes the form:

$$x_{k+1} = x_k - P_k B_k^{-1}\nabla f(x_k). \tag{10}$$

We initialize $P_k^0$ as the identity matrix $I$ and then perform hypergradient descent on $P_k$ using the gradient of $f(x_{k+1})$ with respect to $P_k^i$ to obtain $P_k^{i+1}$:

$$P_k^{i+1} = P_k^i - \eta \frac{\partial f(x_k - P_k^i B_k^{-1}\nabla f(x_k))}{\partial P_k^i}, \tag{11}$$

Table 1: Objective value of the least square problem with hypergradient descent (HGD) on $P_k$ for different BFGS iterations.

| HGD (i) BFGS (k) | 1 | 5 | 10 | 20 |
|---|---|---|---|---|
| 1 | 7.52938 | 6.32887 | 5.22274 | 4.32551 |
| 2 | 1.97834 | 1.95869 | 1.93509 | 1.89111 |
| 3 | 0.88499 | 0.88143 | 0.87703 | 0.86839 |
| 4 | 0.44807 | 0.44746 | 0.44670 | 0.44519 |
| 5 | 0.25669 | 0.25658 | 0.25644 | 0.25617 |

where $\eta$ is the step size for the gradient descent on $P_k$. After $T$ iterations, we employ $P_k^T$ in the update rule 10.

We conduct experiments on the least squares problem to assess the effectiveness of hypergradient descent applied to $P_k$. Each BFGS iteration includes 20 steps of hypergradient descent, after which the most recent $P_k$ identified by hypergradient descent is used in BFGS update. Table 1 presents the experimental results, where each row shows the objective value within one BFGS iteration across different hypergradient descent steps. The results demonstrate that while hypergradient descent shows some improvement over standard BFGS, the benefits become increasingly marginal as iterations progress. This implies that finding an effective $P_k$ is inherently challenging.

This observation motivates exploring methods that can provide meaningful improvements without incurring significant computational costs. This leads us to consider a question: Can we leverage the patterns in optimization trajectories to generate effective step sizes directly? In many optimization scenarios, similar patterns of gradients and Hessian approximations may warrant similar step size adjustments. If these patterns could be learned from data, we might be able to bypass the iterative computation entirely. L2O has shown strong potential in capturing complex patterns and relationships, making it suitable for tasks like predicting step sizes based on optimization state features Liu et al. (2023). By leveraging a neural network, L2O could potentially map the current optimization state to CWSS directly. This approach would allow immediate predictions of effective step sizes without iterative refinement. Before detailing our L2O model, we first establish theoretical conditions for CWSS in BFGS to ensure desirable properties like convergence and stability, which will guide our L2O design.

### 3.3 SUFFICIENT CONDITIONS FOR COORDINATE-WISE STEP SIZES WITH THEORETICAL GUARANTEE

Effective CWSS are crucial for ensuring that each BFGS iteration leads towards a solution. Rather than allowing the L2O model to determine these step sizes arbitrarily, which could lead to unpredictable behavior, we aim to unbox this process through theoretical guidance. This section lays the groundwork by identifying sufficient conditions that CWSS must satisfy for provable convergence and stability. By establishing these foundational principles, we provide a systematic basis for constraining and guiding L2O, ensuring adaptive step-size mechanisms enhance BFGS while preserving its desirable characteristics.

To ensure CWSS are theoretically sound and practically beneficial, we propose the following requirements:

1. *(Convergence Guarantee) The generated sequence $x_k$ converges to one of the local minimizers of $f$.*

2. *(Stability Guarantee) Each update moves towards the minimizer.*

3. *(Convergence Rate Guarantee) The method achieves superlinear convergence.*

The first requirement, ensuring the generated sequence converges to a local minimizer, establishes a fundamental guarantee of reliable final outcomes, extending the concept of Fixed Point Encoding Ryu & Yin (2022). The second requirement, instead, shifts focus to the optimization process itself, emphasizing directional accuracy to ensure stable progress by mandating that each update consistently moves towards the minimizer, thereby preventing detours or excessive zigzagging. Finally, the third requirement addresses convergence speed, aiming to preserve the characteristic superlinear

convergence rate of the BFGS method Wright (2006), a key advantage we seek to maintain within our L2O framework.

We now present three theorems that provide sufficient conditions for coordinate-wise step sizes to satisfy the proposed requirements. The proofs are provided in Appendix A.

**Theorem 1.** *Let $\{x_k\}$ be the sequence generated by equation 10. If the coordinate-wise step size $P_k$ satisfies*

$$\|P_k\|_2 \leq \frac{\alpha}{L\|B_k^{-1}\|_2} \tag{12a}$$

$$\|P_k^{-1}\|_2 \leq \frac{\|B_k^{-1}\nabla f(x_k)\|^2}{\beta\nabla f(x_k)^\top B_k^{-1}\nabla f(x_k)} \tag{12b}$$

*for certain $0 < \alpha < 2$ and $\beta > 0$, where $L$ is the Lipschitz constant of gradients and $B_k$ is the approximate Hessian generated by BFGS, then the sequence of gradients converges to zero: $\lim_{k\to\infty}\|\nabla f(x_k)\|_2 = 0$.*

**Remark.** *Theorem 1 establishes sufficient conditions for gradient convergence while maintaining substantial implementation flexibility. The theorem's bounds on $P_k$ are particularly accommodating: the lower bound of its minimal eigenvalue is allowed to be close to zero through appropriate selection of $\beta$, while setting $\alpha$ near 2 allows the upper bound of the maximal eigenvalue to approach $2/(L\|B_k^{-1}\|_2)$, which remains strictly less than 2.*

Theorem 1 suggests a pragmatic simplification: constraining the elements of the coordinate-wise step size to the interval between 0 and 2 should be sufficient for practical implementations. Moreover, theorem 1 indicates that $P_k$ should be computed as a function of both the gradient $\nabla f(x_k)$ and Hessian approximation $B_k$, as evidenced by the presence of both gradient and Hessian information in bounds. Notably, these results extend beyond convex optimization, requiring only L-smoothness of the objective function rather than convexity.

**Theorem 2.** *Let $f : \mathbb{R}^n \to \mathbb{R}$ be a twice continuously differentiable convex function that has Lipschitz continuous gradient with $L > 0$. Let $x^*$ denote the unique minimizer of $f$. Suppose that $\{B_k\}$ is a sequence of approximate Hessians such that they are uniformly lower bounded: $\gamma I \preceq B_k$, for certain constant $\gamma > 0$. Let $\{P_k\}$ be a sequence of diagonal matrices with entries $p_{k,i}$ satisfying:*

$$0 < p_{k,i} \leq \frac{2\gamma}{L}, \tag{13}$$

*Define the iterative sequence $\{x_k\}$ by equation 10. Then, the sequence $\{x_k\}$ satisfies:*

$$\|x_{k+1} - x^*\| \leq \|x_k - x^*\|.$$

**Remark.** *Since $B_k$ captures the average Hessian behavior between consecutive points $x_{k-1}$ and $x_k$, its eigenvalues lie within the bounds of $\nabla^2 f(x)$, yielding $\gamma \leq L$. This relationship reveals that the seemingly restrictive upper bound $\frac{2\gamma}{L}$ for $p_{k,i}$ simplifies to 2. This aligns with Theorem 1, as both theorems suggest the coordinate-wise step sizes should lie within the interval between 0 and 2. However, theorem 2 makes an additional assumption of convexity, which enables a stronger guarantee, i.e., each iteration strictly decreases the distance to the optimum.*

The theorem can be reduced to classical optimization methods in certain scenarios. For instance, setting $B_k = I$ and $P_k = \alpha I$ with $\alpha \leq 2/L$ yields the standard gradient descent method with constant step size. Moreover, the proof of theorem 2 indicates that optimal $P_k$ values should minimize the spectral radius of $T_k = I - P_k B_k^{-1} H_k$. As the algorithm progresses ($k \to \infty$), $B_k$ approaches $H_k$, suggesting that $P_k$ should converge to the identity matrix. This convergence behavior is formally established in the subsequent theorem.

**Theorem 3.** *Let $x^*$ be a local minimizer where the Hessian matrix, $A \equiv \nabla^2 f(x^*)$, is symmetric and positive definite. Assume that the Hessian is Lipschitz continuous in a neighborhood of $x^*$, i.e., there exist constants $K > 0$ and $p \in (0,1]$ such that for all $x$ in this neighborhood: $\|\nabla^2 f(x) - A\| \leq K\|x - x^*\|^p$. If the sequence $\{x_k\}$ generated by equation 10 converges to $x^*$ such that the summability condition $\sum_{k=0}^{\infty}\|x_k - x^*\|^p < \infty$ is satisfied, and if the sequence of coordinate-wise step size matrices $\{P_k\}$ converges to the identity matrix $I$, then $\{x_k\}$ converges to $x^*$ Q-superlinearly.*

Theorem 3 provides crucial insight into the asymptotic behavior of coordinate-wise step sizes. When the iterates are far from the optimum, coordinate-wise step sizes can accelerate convergence by adapting to the local geometry of the objective function. However, as the algorithm approaches the optimum, the BFGS method naturally provides increasingly accurate Hessian approximations. At this stage, additional coordinate-wise scaling becomes unnecessary and could potentially interfere with the superlinear convergence properties of BFGS. It suggests that adaptive schemes for $P_k$ should be designed to gradually reduce their influence as the optimization progresses, eventually allowing the natural BFGS updates to dominate near the optimum.

## 4 L2O MODEL

Building on the theoretical foundations established in the previous section, we now present our L2O model for CWSS tuning in BFGS optimization. Our design is guided by the derived theoretical conditions to ensure stability and convergence, while leveraging neural networks to adapt to the local optimization geometry.

We propose an L2O method using an LSTM (Long Short-Term Memory) network to predict coordinate-wise step sizes Liu et al. (2023). The architecture is structured as follows:

$$h_k, o_k = \text{LSTM}(x_k, \nabla f(x_k), B_k^{-1} \nabla f(x_k), h_{k-1}, \phi_{\text{LSTM}}),$$
$$p_k = \text{MLP}(o_k, \phi_{\text{MLP}}),$$
$$P_k = \text{diag}(2\sigma(p_k)),$$

where, $h_k$ is the LSTM hidden state, initialized randomly for the first iteration, and $o_k$ is the embedding output from the LSTM network. The parameters of the LSTM and MLP (Multi-Layer Perceptron) networks are denoted by $\phi_{\text{LSTM}}$ and $\phi_{\text{MLP}}$, respectively.

A key aspect of our model's design is the enforcement of theoretically-informed bounds on the predicted step sizes. As established in Theorem 1 and 2, specific bounds on $P_k$ are sufficient to guarantee convergence properties. Ideally, these theorems suggest bounds dependent on quantities like the Lipschitz constant $L$ or the Hessian conditioning $\gamma$. However, these parameters are often unknown or computationally prohibitive to estimate accurately during optimization. Consequently, as a practical and robust simplification suggested by the remarks , we constrain the elements of $P_k$ to lie within the interval between 0 and 2. This a deliberate design choice to prioritize robust convergence. By using a scaled sigmoid activation function to enforce range, we compel our L2O agent to operate within a region that is guaranteed to be stable for any function according to our analysis. While this constraint on the sufficient conditions may limit discovering a more aggressive, potentially faster-converging step-size policy, it is a crucial trade-off. Our design explicitly prioritizes stability to prevent catastrophic failures and ensure our method is reliable across a wide range of problems. Within this theoretically-defined safe operating region, the L2O model is then tasked with learning the more nuanced, data-driven strategy for selecting optimal CWSS that accelerate convergence.

To enhance scalability and parameter efficiency, we employ a coordinate-wise LSTM approach, where the same network is shared across all input coordinates, as suggested in Andrychowicz et al. (2016); Lv et al. (2017). This design allows the L2O method to adapt to problems of varying dimensionality without an increase in the number of parameters.

**Training Process** The L2O model is trained on datasets of diverse optimization problems, allowing it to learn common structures and behaviors. The training process involves using the L2O model to solve these problems while simultaneously updating its own parameters. For each training instance, an optimization trajectory is generated starting from a random initial point $x_0$. At each iteration $k$ of this trajectory, the L2O model predicts the step size $P_k$, which is used to compute the next iterate $x_{k+1}$. Immediately after this step, the network's parameters ($\phi_{\text{LSTM}}$ and $\phi_{\text{MLP}}$) are updated via backpropagation based on the resulting objective value $f(x_{k+1})$. This meta-update treats the optimizer's state—including the gradient $\nabla f(x_k)$ and the search direction $B_k^{-1} \nabla f(x_k)$—as fixed inputs. The gradient for the update flows from the loss back through the predicted step size $P_k$ into the network parameters. Unlike many first-order L2O methods that rely on longer unrolling horizons Andrychowicz et al. (2016); Song et al. (2024); Lv et al. (2017), our model updates its parameters after every single optimization step. We deliberately use this frequent, single-step update strategy

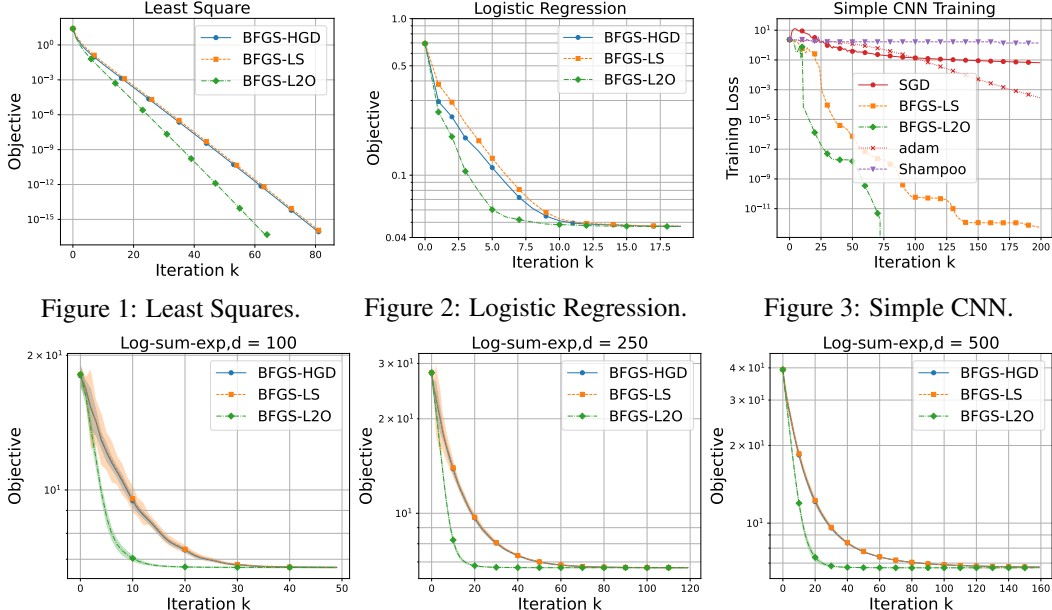

Figure 1: Least Squares.    Figure 2: Logistic Regression.    Figure 3: Simple CNN.

Figure 4: Log-sum-exp functions with different dimensions.

because it is uniquely suited to the quasi-Newton context. In this setting, the step size $P_k$ has a sensitive, dual impact: it simultaneously influences both the next iterate $x_{k+1}$ and the updated Hessian approximation $B_{k+1}$. The immediate feedback provided by single-step updates is critical for the L2O model to effectively learn this complex relationship.

The loss function for training the L2O model is designed to minimize the objective function value at the next iteration, augmented by a regularization term:

$$\min_{\phi_{\text{LSTM}}, \phi_{\text{MLP}}} \mathbb{E}_{f \sim \mathcal{F}}[f(x_{k+1})] + \lambda \|P_k - I\|_F^2 \tag{14}$$

where $\mathcal{F}$ represents the distribution of optimization problems used for training and $\lambda$ is the regularization parameter. The regularization term ensures that as we approach the optimum, the coordinate-wise step sizes converge toward an identity matrix, aligning with the insight from Theorem 3.

In BFGS method, the step size can be viewed as a correction to the Hessian approximation. In early optimization stages, the objective value primarily drives the loss function, and the Hessian approximation may lack precision. Thus, an adaptive CWSS is necessary to enhance the accuracy of the Hessian approximation based on the current state. However, as the optimization nears convergence, the Hessian approximation becomes more accurate, shifting the influence on the loss function to the regularization term. At this point, the CWSS converge to the identity matrix, as further corrections to the Hessian approximation are no longer required.

## 5 EXPERIMENTS

We employed the Adam optimizer as our meta-optimizer to train our L2O model. For classic optimization problems, the training dataset consisted of 32,000 optimization problems with randomly sampled parameters, while a separate test dataset of 1,024 optimization problems was used for evaluation. Our L2O method (BFGS-L2O) was benchmarked against two baselines: Backtracking line search (BFGS-LS) and hypergradient descent (BFGS-HGD). All methods were tuned by experimenting with various parameter settings, and the best-performing configurations were selected for comparison. More details are provided in Appendix B.

**Least Squares Problems**    We first evaluate our L2O method on the classic least squares problems. The objective function is defined as: $\min_x f(x) = \frac{1}{2}\|Ax - b\|^2$, where $A \in \mathbb{R}^{250 \times 500}$ and $b \in \mathbb{R}^{500}$ are randomly generated using a Gaussian distribution.

Figure 1 presents the convergence behavior for the least squares problems, where the optimization process was terminated when the gradient norm fell below $10^{-10}$. As depicted in the figure, BFGS-

Table 2: Wall-clock time analysis for the log-sum-exp problem ($d = 500$).

| Method | Runtime per iteration (s) | Total time to convergence (s) |
|---|---|---|
| **BFGS-L2O (ours)** | 0.109 | **5.79** |
| BFGS-LS | 0.174 | 45.11 |
| BFGS-HGD | 0.089 | 23.90 |

HGD offers a marginal improvement over BFGS-LS. In contrast, our proposed BFGS-L2O method demonstrates a significant reduction in convergence iterations. The nearly linear trajectory (on a log-scale for the objective value) of our BFGS-L2O method is consistent with superlinear convergence.

**Logistic Regression Problems**    Next, we considered logistic regression problems for binary classification. The objective function is given by: $\min_x f(x) = \frac{1}{m} \sum_{i=1}^{m} [b_i \log(h(a_i^T x)) + (1 - b_i) \log(1 - h(a_i^T x))] + \rho \|x\|_2^2$, where $m = 500$, $\{(a_i, b_i) \in \mathbb{R}^{250} \times \{0, 1\}\}_{i=1}^{m}$ are randomly generated, $h(z) = \frac{1}{1+e^{-z}}$ is the sigmoid function.

Figure 2 illustrates the performance on logistic regression problems. While BFGS-HGD achieves a slightly lower objective function value than BFGS-LS during the initial iterations , both baseline methods exhibit similar overall convergence iteration counts, reaching the plateau around 15 iterations. In contrast, our proposed BFGS-L2O method shows notably faster convergence and consistently maintains a lower objective function value throughout the optimization process.

**Log-Sum-Exp Problems**    For the log-sum-exp function, the objective function is: $\min_x f(x) = \log \left( \sum_{i=1}^{m} e^{a_i^T x - b_i} \right)$, where $m = 500$, $\{(a_i, b_i) \in \mathbb{R}^d \times \mathbb{R}\}_{i=1}^{m}$.

Figure 4 displays the results for log-sum-exp problems across different dimensions (d=100,250,500), revealing a clear trend: as dimensionality increases, the performance advantage of our BFGS-L2O method becomes more pronounced. For $d = 500$, BFGS-L2O converges in approximately 40 iterations, while the baselines take around 150-160 iterations—a nearly 4-fold improvement. In addition, our method achieves consistently tighter variance across all dimensions, indicating greater stability. To confirm these gains translate to practical speedups and address the computational overhead of our L2O model, we performed a wall-clock time analysis for the $d = 500$ case. As shown in Table 2, the per-iteration runtime of BFGS-L2O is competitive, confirming the LSTM's overhead is minimal. Crucially, the drastic reduction in iterations results in a total convergence time that is 4.1x faster than BFGS-HGD and 7.8x faster than BFGS-LS. This analysis validates that our method delivers substantial real-world performance gains.

**Simple CNN Training**    To assess the performance of our method on a more complex, non-convex optimization problem, we trained a simple CNN on the MNIST dataset. The detail setting can be found in appendix B. It is worth noting that while second-order methods like BFGS are less commonly employed for training deep neural networks due to computational costs and challenges with sophisticated landscapes, our aim here is specifically to test the adaptability and robustness of BFGS-L2O under such complex, stochastic conditions. To provide a more comprehensive comparison, we benchmarked our method against not only BFGS-LS and SGD but also the widely-used first-order optimizer, Adam Kingma (2015), and a modern quasi-Newton method, Shampoo Gupta et al. (2018). The training loss curves are presented in Figure 3. In this challenging scenario, our proposed BFGS-L2O method demonstrates substantially superior performance. In stark contrast, Shampoo's convergence stalled early at a high loss value, failing to effectively optimize the network on this task. These results highlight the efficiency and stability of our approach, even in a high-dimensional, non-convex setting where traditional and modern baselines struggle.

## 6 CONCLUSIONS

This work investigated the application of coordinate-wise step sizes in the BFGS method. Through theoretical and numerical analyses, we examined the associated benefits and complexities. We rigorously derived sufficient conditions for coordinate-wise step size designed to enhance convergence properties. Building on this theoretical foundation, we developed a L2O model that effectively predicts these step sizes. Experimental results demonstrate that our proposed L2O approach significantly outperforms standard baseline methods in both convergence speed and stability.

**Reproducibility Statement** We have made a concerted effort to ensure the reproducibility of our work. For our theoretical contributions, all key assumptions are explicitly stated in Section 2.2 , and we provide detailed, step-by-step proofs for all theorems in Appendix A. To facilitate the reproduction of our empirical results, Appendix B offers a comprehensive description of the experimental setup. This includes details on the computational environment, L2O model training hyperparameters, the exact procedures for dataset generation, the specific configurations used for all baseline methods, and the complete architecture of the CNN model used in our non-convex experiment.

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

# A PROOFS

## A.1 PROOF OF THEOREM 1

*Proof.* Consider a quadratic function

$$Q(x) = f(y) + \nabla f(y)^\top (x - y) + \frac{\alpha}{2}(x - y)^\top B_k P_k^{-1}(x - y). \tag{15}$$

Basically, we use this quadratic function as a local approximation of $f(x)$ around $y$, and we think the minimum of this quadratic function is a better solution than $y$. This can work only if $Q(x)$ is an overestimation of $f(x)$. Indeed, we can show that:

$$
\begin{aligned}
Q(x) \\
= f(y) + \nabla f(y)^\top (x - y) + \frac{\alpha}{2}(x - y)^\top B_k P_k^{-1}(x - y) \\
\geq f(y) + \nabla f(y)^\top (x - y) + \frac{\alpha}{2}\frac{1}{\|B_k^{-1}\|_2 \|P_k\|_2}\|x - y\|_2^2 \\
\geq f(y) + \nabla f(y)^\top (x - y) + \frac{L}{2}\|x - y\|^2 \\
\geq f(x),
\end{aligned}
$$

where the second inequality uses the condition $\|P_k\|_2 \leq \frac{\alpha}{L\|B_k^{-1}\|_2}$ and the third uses the assumption of $L$-smoothness.

Plugging $x = y - P_k B_k^{-1}\nabla f(y)$ into the inequality, we get the Armijo condition:

$$f(y) - (1 - \frac{\alpha}{2})\nabla f(y)^\top P_k B_k^{-1}\nabla f(y) \geq f(y - P_k B_k^{-1}\nabla f(y)). \tag{16}$$

Let $x_k = y$ and $x_{k+1} = y - P_k B_k^{-1}\nabla f(y)$, we have

$$
\begin{aligned}
f(x_{k+1}) &\leq f(x_k) - (1 - \frac{\alpha}{2})\nabla f(x_k)^\top P_k B_k^{-1}\nabla f(x_k) \\
&\leq f(x_k) - (1 - \frac{\alpha}{2})\beta\frac{(\nabla f(x_k)^\top B_k^{-1}\nabla f(x_k))^2}{\|B_k \nabla f(x_k)\|_2^2} \\
&= f(x_k) - (\beta - \frac{\alpha\beta}{2})\cos^2\theta_k\|\nabla f(x_k)\|_2^2,
\end{aligned}
$$

where $\theta_k$ is the angle between $\nabla f(x_k)$ and $B_k^{-1}\nabla f(x_k)$. In the second inequality, we use the condition that $\lambda_{\max}(P_k) \geq \beta\frac{\nabla f(x_k)^T B_k^{-1}\nabla f(x_k)}{\|B_k^{-1}\nabla f(x_k)\|_2^2}$.

Following the proof of Theorem 3.2 in Wright (2006), summing over all iterations, we have:

$$f(x_{k+1}) \leq f(x_0) - (\beta - \frac{\alpha\beta}{2})\sum_{i=0}^{k}\cos^2\theta_i\|\nabla f(x_i)\|^2. \tag{17}$$

Note that $f(x_0) - f(x_{k+1})$ is lower-bounded by 0 and upper-bounded by $f(x_0)$. Hence when $k$ approaches infinity, we have:

$$\sum_{i=0}^{k}\cos^2\theta_i\|\nabla f(x_i)\|_2^2 < \infty, \tag{18}$$

which implies

$$\cos^2\theta_k\|\nabla f(x_k)\|_2^2 \to 0. \tag{19}$$

Since $\|B_k\|_2\|B_k^{-1}\|_2 < M$,

$$
\begin{aligned}
\cos\theta_k &= \frac{\nabla f(x_k)^\top B_k^{-1}\nabla f(x_k)}{\|\nabla f(x_k)\|_2\|B_k^{-1}\nabla f(x_k)\|_2} \\
&\geq \frac{\lambda_{B_k^{-1}}^{\min}\|\nabla f(x_k)\|_2^2}{\lambda_{B_k^{-1}}^{\max}\|\nabla f(x_k)\|_2^2} \\
&= \frac{1/\|B_k\|_2}{\|B_k^{-1}\|_2} \\
&> \frac{1}{M}.
\end{aligned}
$$

Then we have

$$
\lim_{k\to\infty}\|\nabla f(x_k)\|_2 = 0. \tag{20}
$$

$\square$

### A.2 PROOF OF THEOREM 2

*Proof.* Let $e_k = x_k - x^*$ denote the difference between the current iterate and the minimizer. We can write the update rule as:

$$
e_{k+1} = e_k - P_k B_k^{-1}\nabla f(x_k). \tag{21}
$$

Using the mean value theorem for vector-valued functions, We can write the gradient as:

$$
\begin{aligned}
\nabla f(x_k) &= \nabla f(x_k) - \nabla f(x^*) \\
&= \int_0^1 \nabla^2 f(x^* + te_k)e_k dt \\
&= H_k e_k
\end{aligned}
$$

where:

$$
H_k = \int_0^1 \nabla^2 f(x^* + te_k)dt. \tag{22}
$$

Since $f$ is $L$-smooth , we have:

$$
\nabla^2 f(x) \preceq LI.
$$

Note that $H_k$ is nothing else but the average of the Hessian matrix along the line segment between $x^*$ and $x_k$. Then we have:

$$
H_k \preceq LI.
$$

Substitute $\nabla f(x_k) = H_k e_k$ into the error update, we have:

$$
\begin{aligned}
e_{k+1} &= e_k - P_k B_k^{-1}H_k e_k \\
&= (I - P_k B_k^{-1}H_k)e_k.
\end{aligned}
$$

Consider the matrix $T_k = I - P_k B_k^{-1}H_k$, we will analyze the spectral radius of it. The upper bound of the eigenvalue of $T_k$ is 1, since $P_k$ is a diagonal matrix with positive entries $p_{k,i}$, and $B_k^{-1}$ and $H_k$ are positive definite matrices. The lower bound of the eigenvalue of $T_k$ is:

$$
\begin{aligned}
\lambda_{min}(T_k) &= 1 - \lambda_{max}(P_k)\lambda_{max}(B_k^{-1})\lambda_{max}(H_k) \\
&\geq 1 - \frac{2\gamma}{L}\frac{L}{\gamma} = -1
\end{aligned}
$$

Hence the spectral radius of $T_k$ is less than 1. Then we have:

$$
\|e_{k+1}\|_2 \leq \|T_k\|_2\|e_k\|_2 \leq \|e_k\|_2. \tag{23}
$$

$\square$

A.3 PROOF OF THEOREM 3

*Proof.* The proof is structured as follows. First, we state the necessary and sufficient condition for Q-superlinear convergence from Dennis & Moré (1974), adapted to our update rule. We then decompose this condition into two terms. The remainder of the proof is dedicated to showing that each of these terms converges to zero under our assumptions.

A foundational result from the Theorem 2.2 in Dennis & Moré (1974) states that an iterative method of the form $x_{k+1} = x_k - (B_k^{\text{eff}})^{-1}\nabla f(x_k)$ converges Q-superlinearly to $x^*$ if and only if:

$$\lim_{k \to \infty} \frac{\|(B_k^{\text{eff}} - A)s_k\|}{\|s_k\|} = 0 \tag{24}$$

where $s_k = x_{k+1} - x_k$. In our framework, the update step is given by $s_k = -P_k B_k^{-1}\nabla f(x_k)$. This implies that our effective Hessian is $B_k^{\text{eff}} = B_k P_k^{-1}$. Thus, to prove Q-superlinear convergence, we must demonstrate that:

$$\lim_{k \to \infty} \frac{\|(B_k P_k^{-1} - A)s_k\|}{\|s_k\|} = 0 \tag{25}$$

We add and subtract $B_k$ inside the norm and apply the triangle inequality:

$$\|(B_k P_k^{-1} - A)s_k\| = \|(B_k P_k^{-1} - B_k + B_k - A)s_k\|$$
$$\leq \|B_k(P_k^{-1} - I)s_k\| + \|(B_k - A)s_k\|$$

Dividing by $\|s_k\|$ (for $k$ large enough such that $x_k \neq x^*$), we get:

$$\frac{\|(B_k P_k^{-1} - A)s_k\|}{\|s_k\|} \leq \frac{\|B_k(P_k^{-1} - I)s_k\|}{\|s_k\|} + \frac{\|(B_k - A)s_k\|}{\|s_k\|} \tag{26}$$

The proof of equation 25 reduces to showing that both terms on the right-hand side of equation 26 converge to zero.

The first term is bounded as follows:

$$\frac{\|B_k(P_k^{-1} - I)s_k\|}{\|s_k\|} \leq \|B_k\|\|P_k^{-1} - I\| \tag{27}$$

By the theorem's assumption, $\lim_{k \to \infty} P_k = I$. Since matrix inversion is a continuous operation on the space of invertible matrices, this implies $\lim_{k \to \infty} P_k^{-1} = I^{-1} = I$. Therefore, $\lim_{k \to \infty} \|P_k^{-1} - I\| = 0$.

Furthermore, the sequence $\{\|B_k - A\|_M\}$ converges, as shown in the proof of Proposition 4. This implies that $\{\|B_k - A\|_M\}$ is bounded, and consequently, the sequence of matrix norms $\{\|B_k\|\}$ is also bounded. Since we have the product of a bounded sequence and a sequence converging to zero, the first term converges to zero:

$$\lim_{k \to \infty} \frac{\|B_k(P_k^{-1} - I)s_k\|}{\|s_k\|} = 0 \tag{28}$$

For the second term, we now show that $\lim_{k \to \infty} \frac{\|(B_k - A)s_k\|}{\|s_k\|} = 0$. This relies on the properties of the BFGS update formula itself. We define a weighted matrix norm $\|Q\|_M = \|MQM\|_F$, where $M = A^{-1/2}$ and $\|\cdot\|_F$ is the Frobenius norm. We carry out the proof process by splitting it into several lemmas.

**Lemma 1.** *Let $\bar{E}_k$ be a symmetric matrix and $\bar{s}_k$ be a non-zero vector. Let $\bar{E}_{k+1}^{quad}$ be the updated error matrix from the BFGS formula in the ideal quadratic case (i.e., where $\bar{y}_k = \bar{s}_k$). Then:*

$$\|\bar{E}_{k+1}^{quad}\|_F^2 = \|\bar{E}_k\|_F^2 - \frac{\|\bar{E}_k \bar{s}_k\|^2}{\|\bar{s}_k\|^2} \tag{29}$$

*Proof.* The proof relies on geometric properties of the update. Let $P_s = \frac{\bar{s}_k \bar{s}_k^T}{\|\bar{s}_k\|^2}$ be the orthogonal projector onto the span of $\bar{s}_k$. One can show that the updated error matrix $\bar{E}_{k+1}^{\text{quad}}$ has two key properties: (1) it annihilates the step direction, $\bar{E}_{k+1}^{\text{quad}} \bar{s}_k = 0$, and (2) its action on the subspace orthogonal to $\bar{s}_k$ is the same as the old error matrix, $\bar{E}_{k+1}^{\text{quad}}(I - P_s) = \bar{E}_k(I - P_s)$. Using the Pythagorean theorem for the Frobenius norm, we have:

$$\|\bar{E}_{k+1}^{\text{quad}}\|_F^2 = \|\bar{E}_{k+1}^{\text{quad}} P_s\|_F^2 + \|\bar{E}_{k+1}^{\text{quad}}(I - P_s)\|_F^2$$
$$= 0 + \|\bar{E}_k(I - P_s)\|_F^2 = \|\bar{E}_k\|_F^2 - \|\bar{E}_k P_s\|_F^2$$

The result follows from noting that $\|\bar{E}_k P_s\|_F^2 = \frac{\|\bar{E}_k \bar{s}_k\|^2}{\|\bar{s}_k\|^2}$. □

**Lemma 2.** *Let $\bar{E}_{k+1} = \bar{E}_{k+1}^{quad} + \Delta_k$, where $\Delta_k$ is the perturbation arising from the non-quadratic term $\bar{\epsilon}_k = \bar{y}_k - \bar{s}_k$. Under the theorem's assumptions, for sufficiently large $k$, there exists a constant $C > 0$ such that:*

$$\|\Delta_k\|_F \leq C \frac{\|\bar{\epsilon}_k\|}{\|\bar{s}_k\|} \tag{30}$$

*Proof.* The perturbation is given by $\Delta_k = \bar{E}_{k+1} - \bar{E}_{k+1}^{\text{quad}}$. From the BFGS update formula, this simplifies to:

$$\Delta_k = \frac{\bar{y}_k \bar{y}_k^T}{\bar{y}_k^T \bar{s}_k} - \frac{\bar{s}_k \bar{s}_k^T}{\|\bar{s}_k\|^2} \tag{31}$$

Substitute $\bar{y}_k = \bar{s}_k + \bar{\epsilon}_k$. The denominator becomes $\bar{y}_k^T \bar{s}_k = \|\bar{s}_k\|^2 + \bar{\epsilon}_k^T \bar{s}_k$. As $k \to \infty$, $\|\bar{\epsilon}_k\|/\|\bar{s}_k\| \to 0$, so for large $k$, we have $|\bar{\epsilon}_k^T \bar{s}_k| \leq \|\bar{\epsilon}_k\|\|\bar{s}_k\| \leq \frac{1}{2}\|\bar{s}_k\|^2$. This guarantees the denominator is positive and bounded below by $\frac{1}{2}\|\bar{s}_k\|^2$.

Using a common denominator, the numerator of $\Delta_k$ is $\|\bar{s}_k\|^2(\bar{s}_k + \bar{\epsilon}_k)(\bar{s}_k + \bar{\epsilon}_k)^T - (\|\bar{s}_k\|^2 + \bar{\epsilon}_k^T \bar{s}_k)\bar{s}_k \bar{s}_k^T$. Expanding and simplifying yields:

$$\text{Num}(\Delta_k) = \|\bar{s}_k\|^2(\bar{s}_k \bar{\epsilon}_k^T + \bar{\epsilon}_k \bar{s}_k^T + \bar{\epsilon}_k \bar{\epsilon}_k^T) - (\bar{\epsilon}_k^T \bar{s}_k)\bar{s}_k \bar{s}_k^T \tag{32}$$

Taking the Frobenius norm and using the triangle inequality:

$$\|\text{Num}(\Delta_k)\|_F \leq \|\bar{s}_k\|^2(2\|\bar{s}_k\|\|\bar{\epsilon}_k\| + \|\bar{\epsilon}_k\|^2) + \|\bar{\epsilon}_k\|\|\bar{s}_k\|\|\bar{s}_k\|^2 = 3\|\bar{s}_k\|^3\|\bar{\epsilon}_k\| + \mathcal{O}(\|\bar{\epsilon}_k\|^2)$$

Dividing the bound on the numerator by the lower bound on the denominator gives:

$$\|\Delta_k\|_F \leq \frac{3\|\bar{s}_k\|^3\|\bar{\epsilon}_k\| + \cdots}{\frac{1}{2}\|\bar{s}_k\|^4} = 6\frac{\|\bar{\epsilon}_k\|}{\|\bar{s}_k\|} + \mathcal{O}\left(\left(\frac{\|\bar{\epsilon}_k\|}{\|\bar{s}_k\|}\right)^2\right)$$

Since $\|\bar{\epsilon}_k\|/\|\bar{s}_k\| \to 0$, for some constant $C$, the bound holds. □

**Lemma 3.** *Under the theorem's assumptions, for any symmetric matrix $B_k$, the updated matrix $B_{k+1}$ satisfies:*

$$\|B_{k+1} - A\|_M \leq \left[(1 - \alpha\theta_k^2)^{1/2} + C_1\sigma_k\right]\|B_k - A\|_M + C_2\sigma_k \tag{33}$$

*for some positive constants $C_1, C_2$, where $\sigma_k = \max\{\|x_{k+1} - x^*\|^p, \|x_k - x^*\|^p\}$, $\alpha \in (0, 1]$ is a constant, and*

$$\theta_k = \frac{\|M(B_k - A)s_k\|_F}{\|B_k - A\|_M\|M^{-1}s_k\|_F} \tag{34}$$

*Proof.* We start with the decomposition $\bar{E}_{k+1} = \bar{E}_{k+1}^{\text{quad}} + \Delta_k$ and take norms:

$$\|\bar{E}_{k+1}\|_F \leq \|\bar{E}_{k+1}^{\text{quad}}\|_F + \|\Delta_k\|_F \tag{35}$$

Using Lemma 1, this becomes $\|\bar{E}_{k+1}\|_F \leq (1-\theta_k^2)^{1/2}\|\bar{E}_k\|_F + \|\Delta_k\|_F$. From Lemma 2 and the Lipschitz assumption ($\|\bar{e}_k\|/\|\bar{s}_k\| \leq \text{const} \cdot \sigma_k$), we get:

$$\|\bar{E}_{k+1}\|_F \leq (1-\theta_k^2)^{1/2}\|\bar{E}_k\|_F + C\sigma_k \tag{36}$$

Let $\phi_k = \|\bar{E}_k\|_F = \|B_k - A\|_M$. We have shown $\phi_{k+1} \leq \phi_k + C\sigma_k$. Since $\sum \sigma_k < \infty$, it follows that the sequence $\{\phi_k\}$ is bounded. Let $P$ be an upper bound for $\{\phi_k\}$. We can artificially introduce a $\phi_k$ term. For any $C_1 > 0$:

$$C\sigma_k = C_1\sigma_k\phi_k + (C - C_1\phi_k)\sigma_k \tag{37}$$

The term $(C - C_1\phi_k)$ is bounded, say by $C_2'$, since $\phi_k$ is bounded. So, $(C - C_1\phi_k)\sigma_k \leq C_2'\sigma_k$. Let $C_2 = \max(C, C_2')$.

$$\phi_{k+1} \leq (1-\theta_k^2)^{1/2}\phi_k + C_1\sigma_k\phi_k + C_2\sigma_k = \left((1-\theta_k^2)^{1/2} + C_1\sigma_k\right)\phi_k + C_2\sigma_k \tag{38}$$

The value $\alpha$ from the original lemma statement can be taken as 1. $\qquad\square$

**Lemma 4.** *Let $\{\phi_k\}$ and $\{\delta_k\}$ be sequences of non-negative numbers such that $\phi_{k+1} \leq (1+\delta_k)\phi_k + \delta_k$ and $\sum_{k=0}^{\infty} \delta_k < \infty$. Then the sequence $\{\phi_k\}$ converges.*

*Proof.* We first show that $\{\phi_k\}$ is bounded. Let $\mu_m = \prod_{j=0}^{m-1}(1+\delta_j)$. Since $\sum \delta_j < \infty$, the infinite product $\prod(1+\delta_j)$ converges, which implies that the sequence of partial products $\{\mu_m\}$ is bounded. Let $\mu = \sup_m \mu_m < \infty$. From the given inequality, we have $\phi_{k+1} \leq (1+\delta_k)\phi_k + \delta_k$. Dividing by $\mu_{k+1} = \mu_k(1+\delta_k)$ gives:

$$\frac{\phi_{k+1}}{\mu_{k+1}} \leq \frac{(1+\delta_k)\phi_k}{\mu_k(1+\delta_k)} + \frac{\delta_k}{\mu_{k+1}} = \frac{\phi_k}{\mu_k} + \frac{\delta_k}{\mu_{k+1}} \tag{39}$$

Since $\mu_k \geq 1$ for all $k$, we have $\mu_{k+1} \geq 1$, and thus $\frac{\delta_k}{\mu_{k+1}} \leq \delta_k$. This yields:

$$\frac{\phi_{k+1}}{\mu_{k+1}} \leq \frac{\phi_k}{\mu_k} + \delta_k \tag{40}$$

Summing this relation from $k = 0$ to $m-1$:

$$\frac{\phi_m}{\mu_m} \leq \frac{\phi_0}{\mu_0} + \sum_{k=0}^{m-1} \delta_k \tag{41}$$

Since $\sum \delta_k < \infty$, the right-hand side is bounded by some constant $C$. Thus, $\phi_m \leq \mu_m C \leq \mu C$, which shows that $\{\phi_k\}$ is a bounded sequence.

Now, we show that $\{\phi_k\}$ has a unique limit point. Let $\phi_{inf} = \liminf_{k\to\infty} \phi_k$ and $\phi_{sup} = \limsup_{k\to\infty} \phi_k$. By definition, $\phi_{inf} \leq \phi_{sup}$. We must show $\phi_{sup} \leq \phi_{inf}$. Unfolding the recurrence for $j$ steps from an index $k$ gives:

$$\phi_{k+j} \leq \phi_k \prod_{i=k}^{k+j-1}(1+\delta_i) + \sum_{i=k}^{k+j-1} \delta_i \prod_{l=i+1}^{k+j-1}(1+\delta_l) \tag{42}$$

Taking the limit superior as $j \to \infty$ on both sides:

$$\phi_{sup} \leq \phi_k \prod_{i=k}^{\infty}(1+\delta_i) + \sum_{i=k}^{\infty} \delta_i \prod_{l=i+1}^{\infty}(1+\delta_l) \tag{43}$$

This inequality holds for all $k$. Now, we take the limit inferior as $k \to \infty$. Since $\sum \delta_k < \infty$, we have $\lim_{k\to\infty} \prod_{i=k}^{\infty}(1+\delta_i) = 1$ and $\lim_{k\to\infty} \sum_{i=k}^{\infty} \delta_i = 0$. This yields:

$$\phi_{sup} \leq (\liminf_{k\to\infty} \phi_k) \cdot 1 + 0 = \phi_{inf} \tag{44}$$

Since $\phi_{sup} \leq \phi_{inf}$ and $\phi_{inf} \leq \phi_{sup}$, we must have $\phi_{sup} = \phi_{inf}$. Therefore, the sequence $\{\phi_k\}$ converges. $\qquad\square$

**Proposition 4.** *Under the theorem's assumptions,* $\lim_{k\to\infty} \frac{\|(B_k-A)s_k\|}{\|s_k\|} = 0$.

*Proof.* Let $\phi_k = \|B_k - A\|_M$. The inequality from Lemma 3 is of the form $\phi_{k+1} \leq (1+C_1\sigma_k)\phi_k + C_2\sigma_k$. Let $\delta_k = \max\{C_1\sigma_k, C_2\sigma_k\}$. Since $\sum \sigma_k < \infty$, we have $\sum \delta_k < \infty$. The inequality can be written as $\phi_{k+1} \leq (1 + \delta_k)\phi_k + \delta_k$. By Lemma 4, we conclude that the sequence of error norms $\{\|B_k - A\|_M\}$ converges.

Using the inequality $(1 - x)^{1/2} \leq 1 - x/2$ for $x \in [0, 1]$, Lemma 3 gives:

$$\|B_{k+1} - A\|_M \leq \left(1 - \frac{\alpha\theta_k^2}{2}\right) \|B_k - A\|_M + \mathcal{O}(\sigma_k) \tag{45}$$

Rearranging the terms, we have:

$$\frac{\alpha\theta_k^2}{2}\|B_k - A\|_M \leq (\|B_k - A\|_M - \|B_{k+1} - A\|_M) + \mathcal{O}(\sigma_k) \tag{46}$$

Summing both sides from $k = 0$ to $N$:

$$\frac{\alpha}{2}\sum_{k=0}^{N}\theta_k^2\|B_k - A\|_M \leq (\|B_0 - A\|_M - \|B_{N+1} - A\|_M) + \sum_{k=0}^{N}\mathcal{O}(\sigma_k) \tag{47}$$

As $N \to \infty$, the right-hand side is bounded because $\{\|B_k - A\|_M\}$ converges and $\sum \mathcal{O}(\sigma_k)$ is finite. Thus, the series on the left must converge:

$$\sum_{k=0}^{\infty}\theta_k^2\|B_k - A\|_M < \infty \tag{48}$$

Since the series converges, its general term must approach zero: $\lim_{k\to\infty}\theta_k^2\|B_k - A\|_M = 0$. Let $\mathbb{L} = \lim_{k\to\infty}\|B_k - A\|_M$. We consider two cases for $\mathbb{L}$.

- **Case 1: $\mathbb{L} > 0$.** Since the sequence $\{\|B_k - A\|_M\}$ converges to a positive number, it is bounded away from zero for large $k$. For the product $\theta_k^2\|B_k - A\|_M$ to converge to zero, we must have $\lim_{k\to\infty}\theta_k^2 = 0$, which implies $\lim_{k\to\infty}\theta_k = 0$.

- **Case 2: $\mathbb{L} = 0$.** In this case, $\lim_{k\to\infty}\|B_k - A\|_M = 0$.

From the definition of $\theta_k$, we have $\|M(B_k - A)s_k\|_F = \theta_k \cdot \|B_k - A\|_M \cdot \|M^{-1}s_k\|_F$. By equivalence of norms in finite-dimensional spaces, there exists a constant $C$ such that:

$$\|(B_k - A)s_k\| \leq C \cdot \theta_k \cdot \|B_k - A\|_M \cdot \|s_k\|$$

Dividing by $\|s_k\|$:

$$\frac{\|(B_k - A)s_k\|}{\|s_k\|} \leq C \cdot \theta_k \cdot \|B_k - A\|_M$$

We analyze the limit of the right-hand side. In Case 1 ($\mathbb{L} > 0$), we have $\theta_k \to 0$ and $\|B_k - A\|_M \to \mathbb{L}$ (bounded). Thus the product converges to zero. In Case 2 ($L = 0$), we have $\|B_k - A\|_M \to 0$ and $\theta_k$ is bounded (as $0 \leq \theta_k \leq 1$). Thus the product also converges to zero. In both possible cases, the right-hand side converges to zero, which proves the proposition. $\square$

Finally, we have shown that both terms on the right-hand side of inequality equation 26 converge to zero as $k \to \infty$. This implies that the condition equation 25 is satisfied. Therefore, the sequence $\{x_k\}$ converges to $x^*$ Q-superlinearly. $\square$

## B  DETAILED EXPERIMENTAL SETUP

This section provides supplementary details regarding our experimental settings, including the computational environment, dataset generation procedures, L2O model training, baseline configurations, and the specifics of the neural network training task.

## B.1 Computational Environment

Our experiments were conducted using Python 3.9 and PyTorch 1.12. The underlying system was Ubuntu 18.04, equipped with an Intel Xeon Gold 5320 CPU and two NVIDIA RTX 3090 GPUs.

## B.2 L2O Model Training (BFGS-L2O)

The L2O parameters for our BFGS-L2O model were trained using Adam as the meta-optimizer. The Adam learning rate was set to $1 \times 10^{-3}$, and we processed a batch size of 64 optimization problems for each update. The L2O model underwent a total of 200 such training updates.

## B.3 Datasets for Classic Optimization Problems

For the classic optimization problems (Least Squares and Log-Sum-Exp), the L2O training dataset consisted of 32,000 randomly generated problem instances. A separate test dataset of 1,024 instances was used for evaluation. Specific parameter generation for each problem type is detailed below.

**Least Squares Problem**   The objective function is:

$$\min_x f(x) = \frac{1}{2}|Ax - b|^2,$$

where $A \in \mathbb{R}^{m \times n}$ and $b \in \mathbb{R}^n$. For our experiments, we used m=250 and n=500. The elements of A and b were randomly generated using a Gaussian distribution. Following the setup in Liu et al. (2023) (from your main text), sparsity was introduced into A by setting 90% of its elements to zero.

**Log-Sum-Exp Problem**   The objective function is:

$$\min_x f(x) = \log\left(\sum_{i=1}^{m} e^{a_i^T x - b_i}\right),$$

where $m = 500$ (number of exponential terms). The vectors $\{(a_i, b_i) \in \mathbb{R}^d \times \mathbb{R}\}_{i=1}^m$ were generated following the dataset generation process described in Rodomanov & Nesterov (2021) to ensure the optimal solution is $x^* = 0$. We first generate auxiliary random vectors $\{\hat{a}_i\}_{i=1}^m$ by sampling uniformly from the interval $[0, 1]$. We then generate $\{b_i\}_{i=1}^m$ from the standard normal distribution. Using these, we define an auxiliary function $\hat{f}(x) = \log\left(\sum_{i=1}^m e^{\hat{a}_i^T x - b_i}\right)$. Finally, we set $a_i = \hat{a}_i - \nabla\hat{f}(0)$, ensuring that the optimal solution of $f(x)$ is 0.

## B.4 Baseline Method Configurations

- BFGS-LS (BFGS with Backtracking Line Search): The step size was initialized to 1 at each iteration. The backtracking line search iteratively scaled the step size by 0.8 until the Armijo condition $f(x_k + \alpha_k d_k) \le f(x_k) + c_1 \alpha_k \nabla f(x_k)^T d_k$ is satisfied. Here $d_k$ is the descent direction and $c_1 = 10^{-4}$.
- BFGS-HGD (BFGS with Hypergradient Descent): Within each BFGS iteration, coordinate-wise step sizes $P_k$ were initialized as the identity matrix and then refined by performing 20 iterations of hypergradient descent with hyper step size $\eta = 10^{-2}$.

## B.5 Simple CNN Training Details

The Convolutional Neural Network (CNN) used for the MNIST dataset experiments processes input images of size $28 \times 28 \times 1$. The architecture begins with a first convolutional layer applying 2 filters with a $3 \times 3$ kernel, stride 1, and padding 1, followed by a ReLU activation, resulting in a $28 \times 28 \times 2$ volume. This is then downsampled by a $2 \times 2$ max pooling layer with a stride of 2, producing a $14 \times 14 \times 2$ volume. A second convolutional layer follows, applying 3 filters with a $3 \times 3$ kernel, stride 1, and padding 1, again followed by ReLU activation, yielding a $14 \times 14 \times 3$ volume. This is further downsampled by a second $2 \times 2$ max pooling layer with a stride of 2, resulting in a $7 \times 7 \times 3$ volume. This output is then flattened into a vector of 147 features, which feeds into a fully connected layer that produces 10 output units, corresponding to the logits for the 10 MNIST classes.

## C  GAIN OF CWSS

To illustrate the potential benefits of CWSS in the BFGS method, let us consider the theoretical implications of relaxing the constraint that step size should be a scalar. Assume we have identified an optimal scalar step size, denoted by $\alpha_k^*$, for the $k$-th iteration. Since the restriction of a convex function to a line remains convex Boyd & Vandenberghe (2004), this optimal step size $\alpha_k^*$ satisfies:

$$
\frac{d}{d\alpha_k}f(x_{k+1})\bigg|_{\alpha_k=\alpha_k^*}
$$
$$
=\frac{d}{d\alpha_k^*}f(x_k-\alpha_k^*B_k^{-1}\nabla f(x_k)) \tag{49}
$$
$$
=-\nabla f(x_k-\alpha_k^*B_k^{-1}\nabla f(x_k))^\top B_k^{-1}\nabla f(x_k)
$$
$$
=0.
$$

When constrained to a scalar form, $\alpha_k^*$ guarantees optimality along the single search direction $B_k^{-1}\nabla f(x_k)$. However, if we allow the step size to be a diagonal matrix $P_k$ rather than a scalar, the optimality condition of $\alpha_k^*$ may no longer hold. By extending to a coordinate-wise approach, we aim to further minimize the objective function by adjusting each coordinate independently, which can potentially achieve a lower function value than with $\alpha_k^*$ alone. To explore this, let $P_k = \alpha_k^* I$, and consider the partial derivative of $f(x_{k+1})$ with respect to $P_k$ at this point:

$$
\frac{\partial}{\partial P_k}f(x_k-P_kB_k^{-1}\nabla f(x_k))\bigg|_{P_k=\alpha_k^*I} = \tag{50}
$$
$$
\text{diag}(-\nabla f(x_k-\alpha_k^*B_k^{-1}\nabla f(x_k))\odot B_k^{-1}\nabla f(x_k)),
$$

where $\odot$ denotes the Hadamard (element-wise) product. Since $B_k^{-1}\nabla f(x_k) \neq 0$, the derivative in equation 50 equals zero only if $\nabla f(x_k-\alpha_k^*B_k^{-1}\nabla f(x_k)) = 0$. Since the optimum does not generally lie on the direction of $B_k^{-1}\nabla f(x_k)$, the dot product being zero in equation 49 does not imply that the Hadamard product is also zero in equation 50. This observation suggests that even we know the optimal scalar step size $\alpha^*$, we can still find coordinate-wise step sizes that could achieve a more effective descent. To determine suitable coordinate-wise step sizes, we employ hypergradient descent. Defining $g(p) = f(x_k - p \odot B_k^{-1}\nabla f(x_k))$, where $p$ is the diagonal of $P$, we can analyze the smoothness of $g(p)$ as follows:

$$
\|\nabla g(p_1) - \nabla g(p_2)\|
$$
$$
= \|(\nabla f(x_k - p_1 \odot B_k^{-1}\nabla f(x_k))
$$
$$
- \nabla f(x_k - p_2 \odot B_k^{-1}\nabla f(x_k))\|
$$
$$
\leq L\|(p_1-p_2)\odot B_k^{-1}\nabla f(x_k)\|
$$
$$
\leq L\|B_k^{-1}\nabla f(x_k)\|\|p_1-p_2\|
$$
$$
\leq LR\|p_1-p_2\|,
$$

where $L$ is the Lipschitz constant of $\nabla f$ and $R$ is from assumption 4. This shows that $g(p)$ is $LR$-smooth. To explore this, we can set the coordinate-wise step sizes $P_k$ as:

$$
P_k = \alpha_k^* I - \frac{1}{LR}v_k B_k^{-1}\nabla f(x_k), \tag{51}
$$

where $v_k = \text{diag}(\nabla f(x_k - \alpha_k^* B_k^{-1}\nabla f(x_k)))$, $L$ is the Lipschitz constant of $\nabla f$ and $R$ is from assumption 4. This coordinate-wise step size $P_k$ is theoretically guaranteed to perform better than the scalar step size $\alpha_k^*$:

$$
f(x_k - P_k B_k^{-1}\nabla f(x_k)) \leq f(x_k - \alpha_k^* B_k^{-1}\nabla f(x_k))
$$
$$
- \frac{1}{2LR}|\nabla f(x_k - \alpha_k^* B_k^{-1}\nabla f(x_k)) \odot B_k^{-1}\nabla f(x_k)|^2. \tag{52}
$$

This demonstrates that CWSS in the BFGS method can yield a more substantial decrease in the objective function than a scalar step size.

## D  LIMITATIONS

**Theory-Practice Gap and Practical Simplifications**  Our derivations rely on standard assumptions, such as the L-smoothness of the objective function and the existence of a well-conditioned Hessian approximation. While common in optimization literature, these assumptions may not hold for all practical scenarios, potentially impacting the direct applicability of our theoretical guarantees. Furthermore, our theory provides bounds for CWSS that depend on problem-specific quantities like Lipschitz constants or Hessian conditioning, which are often computationally infeasible to estimate during optimization. To address this, our practical implementation simplifies these bounds by constraining the learned step sizes to the interval [0, 2]. This creates a "safe operating region" that is guaranteed to be stable. While this design choice prioritizes robust convergence, we acknowledge it might preclude the discovery of a more aggressive, faster-converging step-size policy that could exist outside these established bounds.

**Scalability and Extension to Limited-Memory Methods**  A key limitation is the scalability of our method's backbone. This work is developed for the standard BFGS algorithm, which requires storing and updating a dense Hessian approximation, incurring memory and computational costs of $O(d^2)$ per step, where $d$ is the number of parameters. This makes it prohibitive for extremely large-scale models. A natural direction for future work is to adapt our L2O approach to memory-efficient variants like L-BFGS. However, this extension is non-trivial, as our theoretical guarantees do not directly transfer.

- Theorem 1 and 2 rely on the norms and eigenvalue bounds of the explicit $B_k$ matrix, which is never formed in the matrix-free L-BFGS algorithm.
- Theorem 3 guarantee of superlinear convergence is fundamentally tied to the full-memory BFGS update and does not apply to L-BFGS, which typically exhibits a linear convergence rate.

From a practical standpoint, our L2O model could be heuristically combined with L-BFGS by using the search direction computed by the L-BFGS two-loop recursion as an input. While this is a promising practical direction, it would operate without the rigorous theoretical assurances established in this paper. Therefore, developing a new theoretical framework to guarantee the stability and convergence of a learned CWSS policy for L-BFGS remains a significant and important open problem.

## E  LLM USAGE STATEMENT

We utilized a large language model (LLM) as an assistive tool in the preparation of this manuscript. The LLM's role was strictly limited to improving the clarity and readability of the text, including tasks such as grammar correction, spelling checks, rephrasing for conciseness, and improving sentence structure. The LLM was not used for any core research aspects, such as the ideation of the method, the derivation of theoretical results, the design of experiments, or the analysis of the results. The authors have reviewed all suggested edits and take full responsibility for all content presented in this paper.

