# OpenReview forum: "A Learn-to-Optimize Approach for Coordinate-Wise Step Sizes for Quasi-Newton Methods"
_ICLR.cc/2026/Conference — ICLR 2026 Conference Withdrawn Submission_

### Official Review · Reviewer_R9Es · 2025-10-16

**Soundness:** 2
**Presentation:** 3
**Contribution:** 1
**Rating:** 2
**Confidence:** 4

**Summary:**

This paper introduces a modification of BFGS whereby adaptive coordinate-wise step sizes are multiplied to the BFGS step. These step sizes are learned online during optimization via hypergradient descent on a one-step unrolling of the optimization process. Theorems show that this optimizer converges and the modification to BFGS preserves it's superlinear convergence property. Toy experiments show improved training loss for the same iteration count.

**Strengths:**

- Theorems show that with some assumptions, this optimizer will converge to the optimum.
- The optimizer is well-constructed to fit the assumptions under which the optimizer will converge. (See parameterization of $P_k$ on line 344).
- Experiments show improved learning speed on toy tasks.

**Weaknesses:**

- The memory requirement of the LSTM operating coordinate-wise is extremely inflated compared to common optimizers like SGD, Adam, and Muon, which also scale linearly in the problem dimensionality but with extremely small multiplicative constant. The memory taken by the optimizer will become a major bottleneck that will really hurt if the optimizer is ever to be used on realistic-scale problems. While the method may be novel, I struggle to see how it is practically useful.
- The experiments are not on a large enough scale where training speed matters enough to warrant choosing a better optimizer. The largest training experiment involves < ~2k parameters to train, and is on an extremely tiny CNN with only 2 or 3 channels in some layers. I suspect the aforementioned memory issue to prevent scaling up further to the point where this method can become of any use.
- I suspect the training on the small scale problems converges way too fast for the hypergradient descent to have taught the LSTM anything useful. For the LSTM to learn something useful, the underlying problem needs to be hard enough that it is not solved way faster than the LSTM can be trained.
- Preconditions in the theorems proven feel quite contrived. The lower bounded $B_k$ assumption of Theorem 2 feels like it would require the loss function to be strongly convex, which is not the case in many useful applications of such optimizers (BFGS would suffice). A precondition of the superlinear convergence result in Theorem 3 is that the learned $P_k$ matrices converge to the identity over time, which requires BFGS to produce a good Hessian estimate, which in turn requires the problem to be convex and otherwise well-conditioned too.

**Questions:**

Major questions:
- Is there any way to circumvent the poor memory requirement scaling of this optimizer, or do you have any experiments on problems with millions of parameters or more where you compare to other optimizers on wall clock speed?
- What is the time/memory complexity of storing $B_k$ and computing the product in Equation (10), and how does that compare to other optimizers?
- Is there some smaller-scale problem that other optimizers really struggle to solve that your method can solve? Ideally, a problem where the benefits of your optimizer (e.g., shaving off a few minutes during training) actually makes a big difference to the application?
- Figures 1-3: Can we include SGD and Adam on every plot, and compare on wall-clock time and not just iteration count? I want to see that the increased time complexity doesn't cancel out the speed you gain by picking better steps.
- Table 2: What are the timing results for SGD and Adam, and what hardware do you use to perform the timing?

Minor questions:
- Theorems 1-3 operate on deterministic losses and gradients. Does the optimizer converge on stochastic losses/gradient signals that are convex?
- How is $B_k$ picked for Equation (8) to guarantee Equation (9)? Does any Hessian approximation do the trick, or are there some conditions that need to be met? Are we sticking with the choice of $B_k$ from BFGS?
- On line 341, $h_k$ and $o_k$ are outputted by the network, whose inputs are treated as fixed. But if $h_k$ is never used to predict the step size, then where is the training signal for the weights writing to $h_k$ coming from?
- The paragraph on line 313 claims that Theorem 3 shows that $P_k$ will eventually converge to the identity, but that is not what Theorem 3 claims. Theorem 3 actually takes the written statement as an assumption and proves something else instead. Where is the proof for the claim that $P_k$ will eventually converge to the identity?

Notes:
- The inline citation style looks a little strange, consider using a style that has brackets around the citation.

---

### Official Review · Reviewer_ctb9 · 2025-10-29

**Soundness:** 2
**Presentation:** 3
**Contribution:** 3
**Rating:** 6
**Confidence:** 5

**Summary:**

The paper explores the application of coordinate-wise step sizes (CWSS) in the BFGS quasi-Newton method to enhance optimization performance. It begins with a theoretical analysis, deriving sufficient conditions for CWSS that ensure convergence, stability, and superlinear convergence within the BFGS framework. Building on this foundation, the authors propose a learn-to-optimize (L2O) method that uses an LSTM-based neural network to predict optimal CWSS, leveraging past optimization trajectories while adhering to the derived theoretical guarantees. The model is trained to minimize the objective function value at the next iteration, with a regularization term to maintain stability. Experiments on classic optimization problems (least squares, logistic regression, log-sum-exp) and a simple CNN training task demonstrate that the proposed BFGS-L2O method achieves faster convergence compared to traditional backtracking line search and hypergradient descent methods, with improved stability and scalability as problem dimensionality increases.

**Strengths:**

The paper lays out the coordinate-wise step-size (CWSS) idea, its integration into BFGS, and the L2O training protocol in a way that is easy to follow. Key design choices—hard clipping, spectral regularisation, and the separation of offline meta-training from online deployment—are all motivated up-front.
The convergence‐rate and stability theorems are carefully stated, the assumptions are explicit, and the proofs in the appendix are complete enough to be reproducible.

**Weaknesses:**

1. Again, I suggest adding the pseudocode of the proposed method in your appendix, which makes reproduction much simpler.

2. Would learning a single scalar learning rate be possible? Also, there should be a comparison between such a variant and the full coordinate-wise step size approach.

**Questions:**

Since I reviewed this paper in Neurips 2025, I don't have more questions.

---

### Official Review · Reviewer_jDsW · 2025-10-31

**Soundness:** 3
**Presentation:** 3
**Contribution:** 2
**Rating:** 4
**Confidence:** 3

**Summary:**

This paper studies diagonal / coordinate-wise step sizes for quasi-Newton methods, specifically BFGS. The authors:

1. Derive sufficient conditions under which a diagonal step-size matrix
   $
   P_k = \operatorname{diag}(p_{k,1}, p_{k,2}, \ldots, p_{k,d})
   $
   preserves desirable properties of quasi-Newton methods, such as:
   - convergence,
   - stability (each step moves toward the minimizer),
   - and even asymptotic superlinear convergence.

2. Propose a learn-to-optimize (L2O) policy, implemented as an LSTM-like controller, that predicts these per-coordinate step sizes each iteration.

3. Constrain that learned policy using theory-inspired rules, e.g. bounding the diagonal entries of \(P_k\) to a “safe” range and regularizing toward the identity.

4. Show experiments on convex problems (least squares, logistic regression, log-sum-exp) and one small non-convex setting (a CNN on MNIST). They claim faster convergence than:
   - BFGS with classic line search,
   - a hypergradient-based diagonal tuner,
   - and (on the CNN) even Adam / Shampoo.

**Strengths:**

### 1 Clear motivation
The motivation is reasonable: a single global step size can be overly conservative because it is limited by the most “dangerous” direction. A diagonal matrix
$
P_k
$
lets you shrink only the risky coordinates and keep making larger progress along safe coordinates. The paper even argues (conceptually) that this can produce strictly better objective decrease than using only a scalar line-search step.

### 2 Theory-driven design
The theory section is structured around three goals:
1. **Convergence:** the iterates $x_k$ should approach a minimizer $x^\*$.
2. **Stability / descent toward the minimizer:** each update step should not explode and should move in a “good” direction.
3. **Superlinear convergence rate:** asymptotically, you still want quasi-Newton–style fast local convergence.

To support these, the paper gives sufficient conditions like:
- The BFGS matrix approximation $B_k$ remains positive definite and not too ill-conditioned.
- The objective is (locally) smooth enough.
- The diagonal step-size matrix $P_k$ stays in a bounded, positive range and eventually behaves like the identity.

**Weaknesses:**

### 1 Novelty is oversold
The paper repeatedly positions itself as “the first to investigate coordinate-wise step sizes in quasi-Newton (BFGS) with theory + learned policy.” Conceptually, though, what they are doing is extremely close to two well-known ideas:

1. **Diagonal preconditioning / per-parameter scaling.**
   Scaling each coordinate of the update direction by a learned positive factor is, in spirit, just adaptive diagonal preconditioning. That idea is old in both first-order and quasi-Newton contexts.

2. **Learning per-parameter learning rates.**
   The L2O literature has already explored per-parameter learned step sizes for gradient-based optimizers. Here, the base direction is not raw gradient but
   $
   B_k^{-1} \nabla f(x_k),
   $
   which is BFGS-style. That is a nice extension, but it feels incremental rather than a clean conceptual leap.

So I do not fully buy the “this is a qualitatively new paradigm” tone. It feels more like “import L2O-style per-parameter scaling into BFGS, with some theory constraints.”

### 2 Assumptions behind the theory are strong and not empirically tested
The convergence / stability / superlinear claims rely on assumptions that are quite restrictive. Typical statements in the paper (paraphrased) are along the lines of:

- Assume the problem is (locally) convex and smooth.
- Assume the quasi-Newton matrix $B_k$ is positive definite and stays well-conditioned: not exploding in norm and not collapsing toward singular.
- Assume that, asymptotically, the learned diagonal matrix becomes identity:
  $
  P_k \to I
  \quad\text{as } k \to \infty.
  $

Under those assumptions, you can show things like:
1. Iterates converge.
2. Each update step is “safe” (does not blow you up).
3. You recover Q-superlinear convergence similar to classical BFGS.

The problem is:
- These assumptions are not realistic in general deep learning. In high-dimensional non-convex landscapes, $B_k$ (or its inverse) can become badly conditioned early on.
- The paper never measures in practice whether $B_k$ is well-conditioned in their CNN experiment.
- The paper never plots whether $\|P_k - I\|_F$ actually goes to $0$ near convergence.
- The paper never demonstrates that $P_k$ truly stays in the “safe” range for all coordinates, instead of saturating/clipping.

So the theoretical story is elegant but feels very “if the stars align, then we’re safe.” The experimental section never convinces me those stars actually align outside simple convex problems.

In short: the paper *claims* to bridge theory and practice, but it does not *show evidence* that the theory’s preconditions hold where they claim empirical wins.

### 3 Experimental scope is too weak for a strong claim
Almost all experiments are:
- synthetic convex problems (least squares, logistic regression, log-sum-exp with random data),
- or MNIST with a small CNN.

**Questions:**

1. **Scalability:**
   How, exactly, do you apply the per-coordinate LSTM when $d$ is in the millions? Do you literally run it coordinate-wise, or do you group parameters (e.g., per-layer statistics)? Please provide FLOP/memory overhead versus L-BFGS or Adam on a realistically sized model.

2. **Baselines:**
   Why is there no L-BFGS baseline, and no damped/trust-region quasi-Newton baseline? Those are standard in practice and might already address “instability,” especially in non-convex settings.

3. **Tuning fairness:**
   For the CNN experiment: how hard did you tune Adam and Shampoo (learning rate schedules, momentum, preconditioner block sizes, etc.)? If Shampoo “stalls,” is that because Shampoo is fundamentally worse here, or because it wasn’t tuned comparably?

4. **Asymptotic behavior of $P_k$:**
   Can you plot $\|P_k - I\|_F$ over training time? The superlinear convergence argument depends on $P_k \to I$. Do you actually observe that in practice?

5. **Conditioning of $B_k$:**
   Your theorems assume $B_k$ stays positive definite and well-conditioned. Did you measure the condition number of $B_k$ (or its inverse) during training on the CNN? If not, can you?

---

### Official Review · Reviewer_1X5x · 2025-11-01

**Soundness:** 2
**Presentation:** 2
**Contribution:** 2
**Rating:** 2
**Confidence:** 4

**Summary:**

**Summary**

This paper proposes a learning-to-optimize approach for choosing a diagonal stepsize matrix (preconditioner) on top of the scaling matrix generated by BFGS. The authors propose to adopt LSTM as the learning mechanism and safeguarding the learned stepsize using sufficient conditions for convergence. Some numerical experiments demonstrate the superior performance of the proposed approach.

**Strengths:**

**Strength**

The paper is overall well-written and easy to follow. The efficiency of the approach is validated by the experiments.

**Weaknesses:**

**Weaknesses**

I have several concerns regarding the motivation of the proposed approach and the theoretical results.

1. Motivation of BFGS + diagonal stepsize

   I find it unnatural to incorporate a diagonal stepsize (preconditioner) into BFGS. The scaling matrix in BFGS already serves as a preconditioner. And laying another preconditioner on top of it seems incremental and not well-justified. In particular, the affine invariance property of BFGS seems incompatible with the diagonal stepsize.

2. Theoretical results

   Although I don't think the main contributions of the paper lie on the theoretical side, the paper does not take into account the recent developments of quasi-Newton methods (e.g., [1-6]). In particular, the assumptions **A3** and **A4** make the proposed analyses limited. Note that these conditions are typically enforced using Armijo-Wolfe conditions, which again seem incompatible with diagonal stepsize. Finally, the linear and superlinear convergence results only hold under restrictive assumptions. I would suggest that the authors incorporate the recent developments of quasi-Newton methods into the paper.

3. Insufficient experiments

   The current experiments are rather toy. It would be desirable to test on more benchmarks, such as cuTEST.

Overall, the theoretical results in the paper fail to incorporate recent developments of quasi-Newton methods, and the learning-to-optimize approach in the paper is mostly presented as a heuristic and theoretically less interesting. I find this paper not of sufficient quality to be published at ICLR.

**Questions:**

**Questions**

1. **A3**-**A4** are typically enforced with the Armijo-Wolfe condition. How could you guarantee it holds when the diagonal stepsize is generated by learning-to-optimize?
2. On line 262 and 311, why does each update move towards the minimizer?
3. I find **Section 3.1** "gain of coordinate-wise stepsize" unconvincing. Naturally, extending line-search from a scalar to a diagonal stepsize gives more degrees of freedom and would give more progress. However,  the possible cost of high dimensionality introduced by the diagonal stepsize is not discussed in the paper. Could you elaborate more?
4. The hypergradient descent from **Section 3.2** typically lacks convergence guarantees for smooth convex functions. What if you try the hypergradient descent variant in [8]?

**Minor issues**

1. The references in the paper are not well formatted. Please use \citet and \citep properly.

2. Line 44

   are met => is met.

3. Line 52

   I don't think it's accurate to call BFGS a second-order method.

4. Line 60

   What does "incision" mean here? The cutting plane?

5. Line 61

   To my knowledge, the mechanism of [7] does not approximate the exact Hessian.

6. Line 85

   step size and step-size are inconsistent.

7. Line 88

   by theoretical analysis => by the theoretical analysis.

8. Line 129

   damped => damping.

9. Line 191

   $\alpha^*_k$ is not defined.

**References**

[1] Rodomanov, A., & Nesterov, Y. (2021). Greedy quasi-Newton methods with explicit superlinear convergence. *SIAM Journal on Optimization, 31*(1), 785–811.

[2] Rodomanov, A., & Nesterov, Y. (2021). Rates of superlinear convergence for classical quasi-Newton methods. *Mathematical Programming*.

[3] Rodomanov, A., & Nesterov, Y. (2021). New results on superlinear convergence of classical quasi-Newton methods. *Journal of Optimization Theory and Applications, 188*, 744–769.

[4] Jin, Q., & Mokhtari, A. (2021). Non-asymptotic superlinear convergence of standard quasi-Newton methods. arXiv preprint arXiv:2003.13607.

[5] Jin, Q., Jiang, R., & Mokhtari, A. (2025). Non-asymptotic global convergence analysis of BFGS with the Armijo-Wolfe line search. arXiv preprint arXiv:2404.16731.

[6] Jin, Q., Jiang, R., & Mokhtari, A. (2024). Non-asymptotic global convergence rates of BFGS with exact line search. arXiv preprint arXiv:2404.01267.

[7] Kunstner, F., Sanches Portella, V., Schmidt, M., & Harvey, N. (2023). Searching for optimal per-coordinate step-sizes with multidimensional backtracking. *Advances in Neural Information Processing Systems*, *36*, 2725-2767.

[8] Gao, W., Chu, Y., Ye, Y. &amp; Udell, M.. (2025). Gradient Methods with Online Scaling. Proceedings of Thirty Eighth Conference on Learning Theory, in Proceedings of Machine Learning Research 291:2192-2226

**Details Of Ethics Concerns:**

N/A.

---

### Note · Authors · 2025-11-24

I have read and agree with the venue's withdrawal policy on behalf of myself and my co-authors.